# Facilitating transition from maternity leave to work for working mothers: A self-efficacy intervention study

Akiko Kokubo[1] , Katsuhiko Yoshikawa[2] and Chia-Huei Wu[3,4]

[1]School of Management and Information, University of Shizuoka, Shizuoka, Japan; [2]Graduate School of Leadership and Innovation, Shizenkan University, Tokyo, Japan; [3]Management Department, Leeds University Business School, University of Leeds, Leeds, UK and [4]Department of Medical Research, China Medical University, Taichung, Taiwan

## Research Article

**Keywords:**
maternity leave; work–life balance self-efficacy; managerial self-efficacy; work performance; training

**Corresponding author:**
Katsuhiko Yoshikawa;
Email: katsuhiko.yoshikawa@shizenkan.ac.jp

## Abstract

The return from maternity leave to work is a critical career transition period for working mothers. To help their readaptation to work, we developed and examined a training program for cultivating their work–family balance self-efficacy in a pretest–posttest design and investigated the time-lagged effect of the boosted self-efficacy on their employment attitude and in-role performance after they returned to work. Data were collected from 100 maternity leave takers from 16 companies in Japan before the training (Time 1), immediately after it (Time 2) and 6 months after returning to work (Time 3), and from their supervisors at Time 3. We found that maternity leave takers displayed an increase in work–life balance self-efficacy after the training. We also found that work–life balance self-efficacy after the training (Time 2) predicted the participants' in-role performance (Time 3) reported by their supervisors, but not employment attitude reported by the participants (Time 3). Our study thus offers preliminary evidence supporting the effectiveness of the training program in helping maternity leave takers' readaptation to work, potentially supplementing existing family-friendly policies.

## Impact statement

Although governments and organizations have introduced family-responsive practices, such as maternity leave, to encourage women to continue their careers, there remain significant gender gaps in career progress. Return from maternity leave to work is a critical career transition period in working women's professional lives. Many women consider withdrawing from their professional careers after childbirth, and, even when they decide to come back to work, they go through a significant readjustment process, experiencing major efficacy uncertainties. In this study, we developed a training program for working mothers during maternity leave to help them better prepare for their return to work. We examined the program in Japan, where working mothers experience significant pressure to conform to gender-based social roles and face challenges in continuing their career, owing to the country's cultural traditions. Our study shows that the training program helps boost working mothers' work–family balance self-efficacy, which in turn contributes to better supervisor-rated in-role performance after they return to work. Our study suggests that organizations can supplement their family-friendly policies by providing maternity leave takers with our training program to enhance working mothers' work–family balance self-efficacy and helping working mothers better readapt to work.

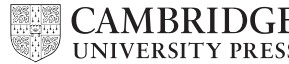

## Introduction

Significant differences between men and women in terms of career progress have been and are still well noted (World Economic Forum, 2017). One of the key reasons for these differences is the gender-based social role expectations that consider women to be primary caregivers in the home and men as breadwinners outside the home (Ridgeway and Correll, 2004). Women tend to dedicate more time to home demands and childcare than men do (OECD, 2012), and their success in an organizational career is more constrained than men's success by home demands (Greenhaus and Callanan, 2013). To encourage women to pursue their careers at work, many governments and organizations have introduced family-responsive policies, such as maternity leave. These policies have been found to decrease job attrition following childbirth (e.g., Glass and Riley, 1998) by helping female employees buffer stress and challenges after childbirth. Nevertheless, the transition from maternity leave to work is a critical career transition in working women's professional lives (Wiese and Heidemeier, 2012). Many women consider 'dropping out of mainstream careers' after having children (Hewlett and Luce, 2005, p. 43), and, even if they choose to return, they go through a significant readjustment process, experiencing major efficacy

uncertainties (Ladge and Greenberg, 2015). We thus argue that organizations should go beyond providing maternity leave and should offer additional resources for female employees to facilitate their adjustment after returning to work.

Drawing on a lens of conservation of resources (Hobfoll, 1989, 2002, 2011), which argues that people seek to protect and gain resources when they face stress, we suggest offering training to female employees during their maternity leave (or upon their return) to boost their personal resources to cope with the transition. Our study thus has two main objectives. First, we develop a case-method training program to boost self-efficacy, a key personal resource for coping with stress (Hobfoll, 2002; Hobfoll et al., 2018) that can be cultivated to cope with a specific situation (Bandura, 1971, 1977, 2001). We focus on work–family balance self-efficacy, a belief in one's ability to succeed in managing the demands of work duties and family obligations (see Cinamon, 2006), because it addresses concerns about the tension between work and family roles, which could prevent employees from pursuing their careers. We examine whether participants displayed an increase in work–family balance self-efficacy after the training in a pretest–posttest design. Second, we examine whether any such self-efficacy boost due to the training can contribute to female employees' adjustment after returning to work. Drawing on the conservation of resources theory (Hobfoll, 1989, 2002, 2011), for its propositions on personal resources in stress coping, and social cognition theory (Bandura, 1971, 1977, 2001), for its propositions on self-efficacy specifically, we propose that enhanced work–family balance self-efficacy can contribute to participants' employment attitudes and in-role performance after they return to the same organizations for work.

We conducted our study in Japan. Japan has one of the lowest levels of female representation in leadership positions among developed countries, with a large proportion of working women leaving their careers after giving birth (Cabinet Office, 2016). The country's societal culture, characterized with collectivism and masculinity, creates strong social pressure for working mothers to conform to gender-based social roles (i.e., being the primary caregiver at home), and workplace norms of long working hours create significant pressure for working mothers in handling both family and home responsibilities (Yoshikawa et al., 2018). These conditions make Japan an ideal location to conduct this study, because working women on maternity leave are more likely to anticipate serious challenges before their counterparts in many other developed countries. Below, we illustrate how we developed a training program based on the principle of self-efficacy development for the first research goal and developed research hypotheses on the effect of work–family balance self-efficacy on employment attitudes and in-role performance for the second research goal.

## Development of the training program for work–family balance self-efficacy

Self-efficacy refers to individuals' belief in their 'capability to exercise some measure of control over their own functioning and over environmental events' (Bandura, 2001, p. 10) or to 'judgments of how well one can execute courses of action required to deal with prospective situations' (Bandura, 1982, p. 122). It has been recognized as a key personal resource for individuals to cope with stress. As indicated by Hobfoll (2002, p. 308), 'those who possessed, for example, high levels of self-efficacy might be more capable of

selecting, altering, and implementing their other resources to meet stressful demands'. Empirically, in work settings self-efficacy is negatively related to job burnout (Shoji et al., 2016) and job strain (Jex and Gudanowski, 1992), helps weaken the relationships between stressors and job strain (Jex and Bliese, 1999) and enables individuals to better use job resources such as job control (Schaubroeck and MerrittSource, 1997) or supervisor social support (Stetz et al., 2006) to cope with job demands.

Self-efficacy is malleable and can be cultivated by four principal sources: mastery experiences, social modeling (or vicarious experiences), verbal persuasion and physical and emotional states (Bandura, 1977, 1982). Mastery experiences provide a direct source of efficacy information because they inform individuals about whether they can perform a behavior well enough to achieve its intended outcomes. Social modeling helps individuals develop their self-efficacy by providing a behavioral script for them to know how to deal with a specific situation effectively. Verbal persuasion helps enhance self-efficacy by reminding individuals that they have the capability to take actions that can achieve their intended outcomes. Finally, physical and emotional states influence self-efficacy because they inform individuals about whether they have the physiological and psychological energy to take actions to make things happen. To date, various training programs have been developed to enhance employees' self-efficacy by strengthening experiential sources of efficacy (e.g., Gist et al., 1989; Eden and Aviram, 1993; McNatt and Judge, 2008; Hahn et al., 2011). We now turn to how we developed a training program to cultivate working mothers' work–family balance self-efficacy by strengthening the four principal sources of self-efficacy in the following aspects: content of materials, activities during the training and time scheduling for the training.

### Content of materials

The first author, who received formal training to develop and organize case-based teaching sessions and has extensive experience interacting with working mothers and their supervisors at various organizations, prepared four case scenarios. Each scenario featured a working mother as the central character and described the challenges that working mothers and the people surrounding them face at work. Prior to conducting this intervention, we ran a series of pilot sessions to test and polish the case materials and instruction. Please see Table 1 for the contents and guiding questions for each case scenario. We developed each case by focusing on specific interaction targets, including a human resources manager, a supervisor, colleagues or customers. We used these four because they are key contacts at work after leave takers return and they involve different aspects of work and resources. For example, whereas concerns about contracts and formal work arrangements are likely topics for discussion with a human resources manager, how one's work is allocated and coordinated is likely determined by supervisors and colleagues on the same team, and how and when individuals deliver a service is directly influenced by who they serve within or outside an organization.

Each case was designed to achieve specific learning objectives. For example, the case that involved the human resources manager incorporated taking the perspective of organizational expectations when negotiating arrangements for a return instead of just asserting the working mothers' own needs for organizational support. The case involving a supervisor was designed for participants to learn how to help a supervisor understand their family constraints and career aspirations at work beyond stereotypical

**Table 1.** Descriptions of case scenarios

| Case | Situation | Other person whose perspective participants are encouraged to consider | Intended learning objectives |
|---|---|---|---|
| 1 | Haruka holds a meeting with the HR manager to discuss conditions for her return to work after maternity leave. However, she struggles to have a constructive conversation with the manager due to her concerns over her ability to manage work and life as well as the company's willingness to support her as a working mother | HR manager | Consider perspectives of HR manager in negotiating work arrangements |
| 2 | Yumi returns to work with an aspiration to advance her career. However, her supervisor is hesitant to assign her more responsibilities because of his stereotypical view of working mothers and the potential negative impact on team performance | Supervisor | Promote mutual understanding with the supervisor to gain desirable work assignments |
| 3 | While working full time, Misaki cannot take overtime work owing to her family responsibilities. This causes communication problems with her colleagues, who often engage with overtime work. The situation deteriorates when her son suddenly becomes ill, and his nursery asks her to keep him at home | Colleagues | Collaboratively coordinate work procedures and protocols with colleagues |
| 4 | Kotomi returns to her sales team after maternity leave. As the sales manager reallocates her customers from other sales representatives, who are mostly male, they start expressing frustrations of perceived favoritism toward Kotomi. She considers proposing alternative work arrangements to mitigate frustrations from colleagues while ensuring customer service quality | Customers | Proactively rearrange the team's procedures, protocols and work assignments, considering its stakeholders |

views about working mothers and the need to understand the supervisor's considerations in deciding on task assignments such as the team's goal and resource constraints. In the case involving colleagues, we sought to enhance participants' ability to collaboratively coordinate work procedures and protocols so that their family-related constraints (e.g., the need to leave the office early) would not obstruct their colleagues from performing tasks smoothly. Finally, the case involving customers was aimed at helping participants learn they could avoid undermining their family-related constraints or even promote their work group's ability to satisfy customers by proactively proposing the rearrangement of work procedures, protocols and task assignments to the team and supervisor, considering expectations from the team's stakeholders (in this case, customers).

Together, these four cases provide opportunities for working mothers to enhance their understanding of resources and approaches for them to manage boundaries between their work and family while making professional contributions to their workplace. Because taking perspectives aids in negotiation and conflict resolution (e.g., Neale and Bazerman, 1982; Galinsky and Mussweiler, 2001; Galinsky et al., 2008), the coverage of four interaction targets should help our participants find better ways to communicate and resolve conflicts with different interaction targets after they return to work. Our materials are designed to enhance participants' work–family balance self-efficacy and managerial self-efficacy in particular. Regarding work–family balance self-efficacy, each case encouraged participants to explore the ways to manage work tasks and relationships without compromising their responsibilities as mothers. Furthermore, the facilitator introduced various possible approaches and provided feedback on the participants' ideas, citing real examples of working mothers from her extensive experience of interacting with them. In terms of managerial self-efficacy, participants also take the perspectives of managers in each case and virtually experience what managers consider when designing job assignments, arranging work procedures and protocols and judging priorities at work. Furthermore, the facilitator brings real concerns

and considerations of managers, drawing from her experience working with supervisors of working mothers as well as teaching managers more broadly.

### Activities during the training

Participants were asked to attend four 2-hour sessions, with one case discussed in each session. Each session usually had 10–25 participants, contingent upon participant availability. They were randomly assigned to groups of three to five upon arrival at the training venue at each session. The first author, who wrote the cases, instructed all the sessions. The instructor first guided participants to discuss what they might do to address the situation as the working mother in the case, and then to take the perspective of the interaction target (e.g., manager). This was to extend the participants' views beyond their own perspective to realize potential opportunities that they could utilize to perform at work while managing their family responsibilities. Further, we distributed supplementary materials, which described the views from the human resource manager and supervisor, respectively, in the first and second sessions to encourage participants to acknowledge different perspectives. Finally, the instructor provided recommended solutions for the case situation and summarized the key perspectives and approaches that participants might adopt in accordance with their own contexts.

The design of the activities offered participants mastery experiences because they could work on a real problem and discover solutions to it during the training session, as well as social modeling by offering to participants recommended solutions based on successful cases. In the meantime, the instructor could deliver verbal persuasion during the training session in several ways, such as by reminding participants to be aware of resources they could use to address challenges if they were in similar situations. Finally, activities in each session were based on group discussion, idea exchange and mutual support, because supportive experiences among participants help sustain their spirits and energy to

overcome potential stressful challenges ahead. We sought to cover and strengthen the four sources of self-efficacy together with participants in each training session because 'the more dependable the experiential sources, the greater are the changes in perceived self-efficacy' (Bandura, 1977, p. 191).

### Training schedule

We delivered four different sessions for a complete cycle of training for several reasons. First, because the four cases addressed different work aspects and relationships, sufficient time for thinking and discussion was needed for each case. It was thus desirable to focus on one case at one time. Second, though malleable, self-efficacy is unlikely to be enhanced sustainably with just one training session. Spreading four sessions across one and a half to 2 months would allow participants to reflect and continually build their capacities to tackle foreseeable challenges; a similar intervention design was used in a prior intervention study (Akkermans et al., 2015). Third, participants must be highly engaged, through thinking, sharing and reflecting, to derive more benefit from training. In essence, self-efficacy is developed via social learning, which cannot exist without engaging and effective social interactions (Bandura, 1971, 1977, 2001). Because participants need time to become familiar with training activities, having multiple sessions is desirable for them to adapt to a training context and become fully engaged during later sessions. Altogether, the design of materials, activities and training schedule is aimed to promote participants' work–family balance self-efficacy. We thus propose:

**Hypothesis 1:** The training program increases participants' (a) work–family balance self-efficacy and (b) managerial self-efficacy.

### Effects of work–family balance self-efficacy on employment attitude and in-role performance after returning to work

Drawing on the conservation of resources theory (Hobfoll, 1989, 2002, 2011) and social cognition theory (Bandura, 1971, 1977, 2001), we expected that work–family balance self-efficacy would, upon working mothers' returning to work, be associated with a stronger attitude to continue their employment and better in-role performance after returning to work.

Conservation of resources theory posits that 'people seek to obtain, retain, and protect resources and that stress occurs when resources are threatened with loss or lost or when individuals fail to gain resources after substantive resource investment' (Hobfoll, 2002, p. 312). Working mothers who are in the transition from maternity leave to work are likely to anticipate loss in their time, effort, attention and energy for delivering the expected performance after returning to work owing to the new challenges in childcare and family duties. The theory also posits that, in a resource loss situation, 'the ability to obtain resource gains becomes of increasing importance, providing emotional respite and an increased ability to sustain goal pursuit' (Hobfoll, 2002, p. 312). Following this, we expected that the boost of work–family self-efficacy due to our training program would enhance working mothers' ability to obtain resource gains when managing their work and family roles.

As for self-efficacy specifically, social cognition theory posits that self-efficacy enhances an individual's confidence in goal attainment in a specific domain, motivating them to persevere in pursuit of the goal and perform well (Bandura, 1971, 1977, 2001). In our research context, we expected that the boost of work–family

balance self-efficacy would help working mothers become more confident in playing both work and family roles and more likely to see how they could still pursue their career goals while fulfilling their family roles. They would also be likely to prevent interference between family and job activities, which would not only help them avoid resource depletion but also enable them to concentrate on tasks at work and do their job well. Without such efficacy, working mothers are likely to experience role conflict between their family and work duties, which can deplete their energy (e.g., Demerouti et al., 2016) and further undermine their capacity to play both roles simultaneously. Our reasoning is in line with the existing empirical findings on work–family balance self-efficacy. For example, those with higher work–family balance self-efficacy reported a stronger commitment to their career (Myers and Major, 2017), which is indirect evidence showing the positive function of work–family balance self-efficacy in career perseverance. In addition, work–family balance self-efficacy is negatively associated with anticipated work–family conflict (Cinamon, 2006), and we know that lower work–family conflict is associated with better performance (e.g., Demerouti et al., 2016). We thus propose:

**Hypothesis 2:** Post-training work–family balance self-efficacy is positively related to (a) employment attitude and (b) in-role performance after participants return to work.

## Method

### Participants

After receiving ethics evaluations and obtaining permission from the first author's affiliated university, we recruited participants through 17 Japanese firms, representing a wide range of industries, including finance, pharmaceuticals, food, manufacturing, logistics and information services. We distributed a leaflet about our workshop to these companies' female employees on maternity leave. The leaflet included information about workshops, researchers and the logistical arrangements for the sessions (e.g., schedule, location). Participants voluntarily signed up through a website we set up for this study. In total, 116 working mothers from 16 companies signed up for the workshop. The average age of participants who signed up was 32.7 years (SD = 3.51) and 87.1% had an undergraduate or higher degree. For 61.2% of them, this was their first maternity leave in their career, and 34.5% and 4.3% were in their second and third, respectively.

We also sought to collect data from maternity leave takers who worked for the same companies but did not participate in our training program to examine potential selection bias. A total of 58 leave takers signed up for our survey as nontrained participants.[1] Their average age was 32.6 years and 84.5% of them had an undergraduate or higher degree. For 48.2% of these leave takers, they were on the first maternity leave in their career, and 44.8% and 6.9% were in their second and third, respectively. A series of *t*-tests and chi-squared tests showed no significant differences between trained participants and nontrained leave takers regarding their demographic backgrounds (i.e., age, tenure, work experience, education, number of children and number of maternity leaves they had taken, including the current one). Regarding psychological characteristics, we found no significant difference in proactive personality, although

---

[1]We cannot calculate the response rate because some companies did not disclose the number of working mothers to whom they distributed invitation emails.

nontrained participants reported significantly lower scores for family support (mean = 5.54 for trained participants and 4.91 for nontrained participants, $t = 3.10$, $p < .01$), managerial self-efficacy (3.47 and 3.06, $t = 2.29$, $p < .05$), work–life balance self-efficacy (3.59 and 3.05, $t = 2.87$, $p < .01$) and employment attitude (4.68 and 3.80, $t = 3.90$, $p < .01$), which suggests that there was some selection bias due to our recruitment procedure. We controlled for these demographic and psychological variables in the following analyses to minimize the impact of the selection bias on the findings.

### Procedure

The workshops took place from January to March 2018 and from November 2018 to March 2019. A vast majority of working mothers return from maternity leave to work in April, when most nurseries in Japan admit new children, and thus our workshops took place when participants were preparing to return to work. This means that participants started and completed the workshops during their maternity leave. We provided multiple rounds of four workshop sessions with different start dates, and participants chose one of them when they signed up.

We collected data from participants before the start of the first session (T1) and at the end of the fourth session (T2). The gap between T1 and T2 surveys was roughly 2 months, while there was slight variation between rounds, due to scheduling issues. Among the 116 working mothers who signed up for our study, 100 completed the program, including four sessions and two surveys (completion rate = 72.5%). At T1, we measured control variables, including demographic variables, proactive personality and family support, and key research variables, including work–family balance self-efficacy and employment attitude, and supplementary research variables (see below for explanations), including anticipated work–family conflict and managerial self-efficacy (i.e., a belief in one's ability to successfully pursue and perform a management position) (Van Vianen, 1999).

At T2, we again measured work–family balance self-efficacy to examine the effectiveness of our training program. We also measured anticipated work–family conflict again at T2 as a manipulation check variable to see if our training program had successfully assisted participants in navigating any potential work–family conflict they might encounter after returning to work and reduced their anticipation of conflict. We also measured managerial self-efficacy at T2 because our training program could also boost participants' confidence in taking managerial roles and we sought to measure and control for this effect when we examined the effect work–family balance self-efficacy on outcomes at T3. We confirmed that all completed participants returned to work after T2.

Six months after participants returned to work (T3), we distributed an additional, follow-up survey to those 100 participants and their supervisors. Eighty-one participants (response rate = 81%) and 72 supervisors (response rate = 72%) completed the survey. At this time, participants rated their employment attitude. We asked supervisors to rate the participants' in-role performance based on their observations during the past 3 months. We used the previous 3 months as the time frame for performance rating because we wanted to capture working mothers' performance after they went through the readaptation period, which typically takes a couple of months. We also suspected that a shorter time frame might lead to performance assessment being influenced by particular family-related events, such as sickness of children, that happened shortly before the survey. Figure 1 describes the flow chart of the study procedure.

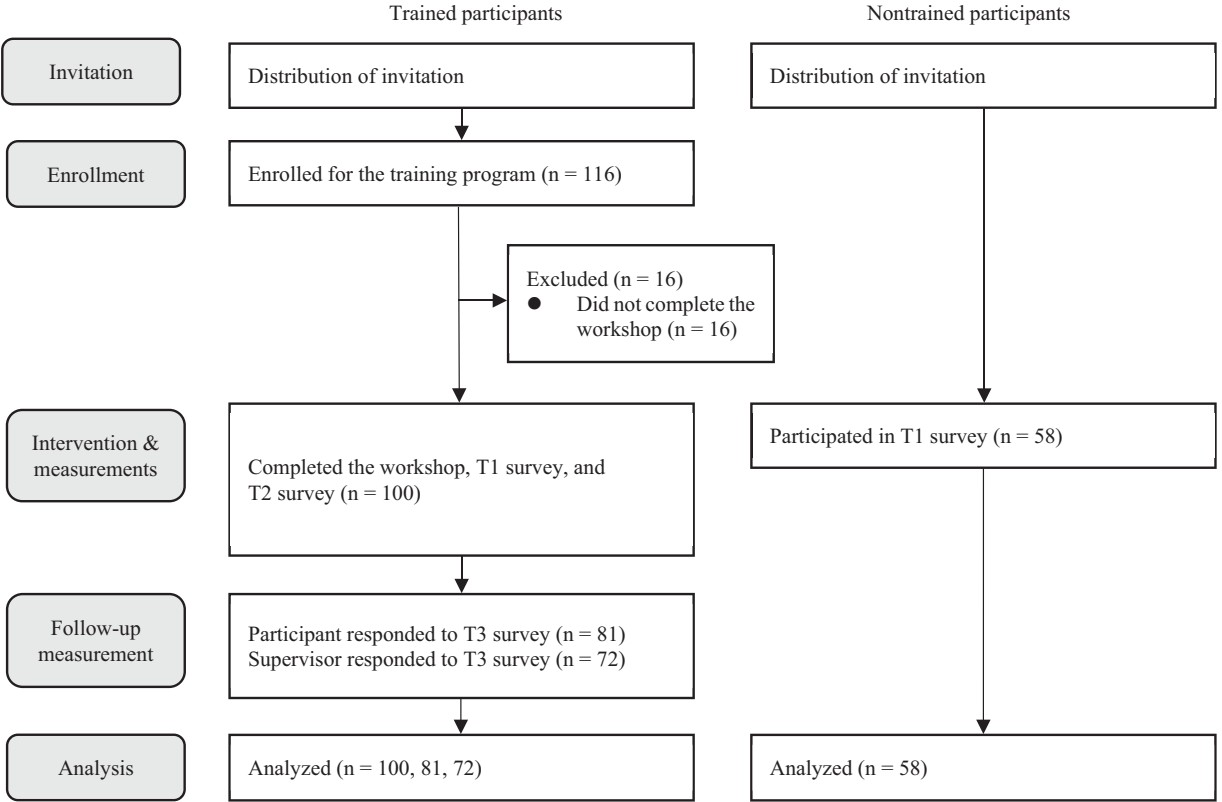

**Figure 1.** Flow chart of the study procedure.

## Measures

Table 2 presents all measures we used. We translated items by the back-translation method (Brislin, 1970). Participants rated these items on a seven-point Likert scale (1 = *strongly disagree*; 7 = *strongly agree*), unless specified otherwise.

### Anticipated work interference with family

We measured this construct at T1 and T2 with a four-item scale by Gutek et al. (1991). We revised the wording, such as friends or personal life, in these items to focus on family life directly.

### Self-efficacy

We measured managerial self-efficacy and work–family balance self-efficacy at T1 and T2. We used Van Vianen's (1999) four-item scale of self-efficacy for managerial jobs to measure managerial self-efficacy. We adapted the same items to measure work–family balance self-efficacy.

### Employment attitude

We measured employment attitude at T1 and T3 using items from Matsui et al.'s (1991) career commitment scale for assessing participants' intentions to continue their employment despite the challenges of family duties.

### In-role performance

We adopted three-item measures of individual task proficiency from Griffin et al. (2007) to capture participants' in-role performance after returning to work. Supervisors were asked to rate in-role performance at T3 using a five-point Likert scale (1 = *not at all*; 5 = *very frequently*).

### Control variables

We also measured several control variables at T1. As for demographic variables, we included participants' age, work experience, tenure in the current organization, education (0 = high school, 1 = college, 2 = undergraduate, 3 = postgraduate), number of children and number of maternity leaves they had taken (including

**Table 2.** Measures at different time points

| Measure | Items | Data collection |
|---|---|---|
| Anticipated work interference with family | After work, I will come home too tired to do some of the things I want to do | T1, T2 |
| | On the job, I will have so much work to do that it takes away from my personal interests regarding my family | |
| | My family/friends will dislike how often I will be preoccupied with my work while I am at home | |
| | My work will take up time that I will want to spend with family/friends | |
| Work–family balance self-efficacy | I expect that I can cope with handling work and family life | T1, T2 |
| | I consider handling work and family life to be very difficult (R) | |
| | Because of my capabilities, I expect I can handle work and family life within a couple of months | |
| | I'm capable of learning the skills for balancing family and work | |
| Managerial self-efficacy | I expect that I can cope with a managerial job | T1, T2 |
| | I consider management positions to be very difficult (R) | |
| | Because of my capabilities, I expect I can get a managerial job within a couple of years | |
| | I'm capable of learning the skills for a managerial job | |
| Employment attitude | Despite the many inconveniences to my family life, I would still like to continue my career | T1, T3 |
| | If my husband dislikes my working, I would quit my job (R) | |
| | If my career cannot coexist with my family life, I will stop working (R) | |
| In-role performance | The employee… Carried out the core parts of her job well | T3 |
| | Completed her core tasks well using the standard procedures | |
| | Ensured her tasks were completed properly | |
| Proactive personality | I am always looking for better ways to do things | T1 |
| | If I believe in an idea, no obstacle will prevent me from making it happen | |
| | If I see something I don't like, I fix it | |
| | No matter what the odds, if I believe in something, I will make it happen | |
| | I excel at identifying opportunities | |
| Family support | My family provided information, suggestions or guidance | T1 |
| | My family gave me tangible assistance | |
| | My family gave me emotional support | |

the current one). Nevertheless, because age and work experience are highly correlated ($r = .97$), as are the number of children and the number of maternity leaves ($r = .90$), we did not control in the analysis for work experience and number of children.

We also controlled for proactive personality, a 'stable tendency to effect environmental change' (Bateman and Crant, 1993, p. 103), which captures individual differences in behavioral tendencies for influencing and changing the environment. Because it has been found to positively relate to self-efficacy (e.g., Wu et al., 2018) and behaviors for changing one's job demands (e.g., Bakker et al., 2012) and making constructive changes at work (e.g., Fuller and Marler, 2009), we wanted to control for its impact when gauging our training effects on self-efficacy and other outcome variables. We used a short-form measure of proactive personality (Parker, 1998; Claes et al., 2005) with five items from Bateman and Crant (1993).

Finally, we controlled for family support, using a three-item scale adopted from Dunkel-Schetter et al. (1987), because participants who have more support from their family are more likely than others to focus on their work and career development (Lirio et al., 2007).

## Results

### Training effects on work–family balance self-efficacy

Table 3 shows descriptive statistics of our variables. We conducted paired $t$-tests ($n = 100$) on T1 and T2 measures of anticipated work interference with family, work–family balance self-efficacy and managerial self-efficacy. We found a significant decrease in anticipated work interference with family (from 4.76 to 4.51; paired $t$-test [$df = 99$] $= -2.97$, $p < .01$), suggesting that our training program helped participants successfully navigate potential challenges they might have experienced after they returned from maternity leave and helped them to reduce concerns about the potential negative impact of their return on their family.

We also found a significant increase of scores in work–family balance self-efficacy (from 3.56 to 3.92; paired $t$-test [$df = 99$] $= 3.27$, $p < .01$), supporting the effectiveness of our training program and hence Hypothesis 1. We also found a significant increase of scores in managerial self-efficacy [from 3.41 to 3.83; paired $t$-test ($df = 99$) $= 5.25$, $p < .01$], suggesting that our training program also boosted participants' confidence in taking managerial roles.

### Effect of work–family balance on employment attitudes and in-role performance

To examine the effects of work–family balance self-efficacy at T2 (immediately after the training) on employment attitudes and in-role performance measured in T3 (6 months after the participants returned to work), we conducted a series of regression analyses.

In these analyses, we also took into account other measures for control (age, tenure, education, number of maternity leaves, proactive personality, family support, work–family balance self-efficacy and employment attitude at T1 and managerial self-efficacy at T1 and T2). Because our sample was nested within 16 companies, we adopted a random-intercept model to account for potential company differences. The largest variance information factor (VIF) value among predictors was 3.52, and the average VIF value was 1.80, suggesting that we did not have strong multicollinearity among predictors. Table 4 summarizes the results. For

both in-role performance and employment attitude, we first included control variables, except for work–family balance self-efficacy (Models 1 and 5, respectively) and then added work–family balance self-efficacy at T1 (Models 2 and 6) and at T3 (Models 3 and 7). Work–family balance self-efficacy showed a significant positive effect on in-role performance ($B = .33$, $p < .01$), and a chi-squared test for $\Delta$-2loglikelihood indicated that the inclusion of work–family balance self-efficacy significantly improved the explanatory power of the model ($p < .01$). However, work–family balance self-efficacy did not have a significant effect on employment attitude ($B = .00$, $p > .10$). Hence, Hypothesis 2 is partially supported.

To gauge the influence of control variables, we also performed the same analysis including only work–family balance self-efficacy at T1 and T2. Models 4 and 8 in Table 4 summarize the results. We found that work–family balance self-efficacy at T2 had a significant positive effect on in-role performance ($B = .30$, $p < .01$) but not on employment attitude ($B = .17$, $p > .10$). These results indicate that the inclusion or exclusion of control variables has limited impact on the analysis results for work–family balance self-efficacy.

Finally, for a robustness check, we used path analysis with the Bayesian estimation[2] to estimate all effects in Models 3 and 6 in Table 4, as well as paths from work–family self-efficacy and managerial self-efficacy at T1 to the same variables at T2 simultaneously using completed data of all variables from T1 to T3 ($n = 64$). Posterior predictive $p$-value (.43) was close to .50, indicating a reasonable model fit (Asparouhov and Muthén, 2010). The results were consistent with Models 3 and 6 in Table 4, with work–family balance self-efficacy having a significant positive effect on in-role performance ($B = .42$, 95% credible interval $= [.11, .73]$) but not on employment attitude ($B = -.12$, credible interval $= [-.57, .33]$). We did the same path analysis to estimate all effects in Models 4 and 8, along with a path from work–family balance self-efficacy at T1 to the same variable at T2, simultaneously. We found no material changes in the effects of work–family balance self-efficacy.

## Discussion

Our study offers a training program that can help cultivate the work–family balance self-efficacy and further reveals that work–family balance self-efficacy, but not managerial self-efficacy, can serve as psychological resource to help working mothers perform

---

[2]We conducted the analysis on Mplus 8.3 (Muthén and Muthén, 1998–2017), using default two Markov Chain Monte Carlo chains with Gibbs sampler. To ensure convergence of estimation, we set Mplus to conduct 20,000 iterations and confirmed that PSR values do not increase (Zyphur and Oswald, 2015). Bayesian analysis produces an estimation of probability distribution of parameters (e.g., coefficients), combining the observed data and hypotheses about the probability distributions of parameters based on existing knowledge about the subject matter (Howson and Urbach, 1993). The initial hypotheses about the probability distributions of parameters are called priors, and the estimated probability distributions are called posteriors. We used two types of prior. First, we used noninformative, diffuse priors, following a general recommendation for cases when limited prior knowledge is available and/or researchers decide to eliminate the influence of priors in estimation. Mplus uses N (0, infinity) for parameters of continuous variables as diffuse priors. Second, to account for the potential bias due to using noninformative priors for small-sample analysis (Smid et al., 2020), we used estimated coefficients and standard errors in regression analyses (Models 3 and 6 in Table 4) for work–family balance self-efficacy to specify a normal prior (see Yuan and MacKinnon, 2009) for the respective paths. The results were largely the same, with no material differences. We report the results from the former.

**Table 3.** Descriptive statistics, correlation and Cronbach's alpha

| | | Mean | SD | 1 | 2 | 3 | 4 | 5 | 6 | 7 | 8 | 9 | 10 | 11 | 12 | 13 | 14 | 15 | 16 | 17 |
|---|---|---|---|---|---|---|---|---|---|---|---|---|---|---|---|---|---|---|---|---|
| 1 | Age (years) (T1) | 32.91 | 3.65 | | | | | | | | | | | | | | | | | |
| 2 | Work experience (years) (T1) | 11.01 | 3.88 | .97 | | | | | | | | | | | | | | | | |
| 3 | Tenure (years) (T1) | 8.52 | 3.98 | .36 | .36 | | | | | | | | | | | | | | | |
| 4 | Education (T1) | 1.95 | .46 | −.14 | −.37 | −.09 | | | | | | | | | | | | | | |
| 5 | Number of children (T1) | 1.51 | .58 | −.01 | .02 | .15 | −.13 | | | | | | | | | | | | | |
| 6 | Number of maternity leaves (T1) | 1.46 | .59 | .02 | .05 | .19 | −.14 | .90 | | | | | | | | | | | | |
| 7 | Proactive personality (T1) | 4.77 | .76 | −.07 | −.08 | −.09 | .06 | −.05 | −.08 | (.69) | | | | | | | | | | |
| 8 | Family support (T1) | 5.52 | 1.18 | −.03 | −.03 | −.12 | .02 | −.03 | −.08 | .12 | (.83) | | | | | | | | | |
| 9 | Managerial self-efficacy (T1) | 3.41 | 1.09 | .05 | .02 | .05 | .14 | .09 | .12 | .28 | .19 | (.86) | | | | | | | | |
| 10 | Work–family balance self-efficacy (T1) | 3.56 | 1.18 | −.10 | −.07 | −.22 | −.08 | .05 | .00 | .18 | .23 | .28 | (.92) | | | | | | | |
| 11 | Anticipated work interference with family (T1) | 4.76 | .98 | −.01 | −.03 | .07 | .10 | .05 | .07 | −.04 | −.16 | .02 | −.38 | (.71) | | | | | | |
| 12 | Employment attitude (T1) | 4.77 | 1.34 | .05 | .04 | −.07 | .03 | .11 | .10 | .26 | .22 | .34 | .24 | .01 | (.86) | | | | | |
| 13 | Managerial self-efficacy (T2) | 3.83 | 1.05 | .08 | .04 | .11 | .12 | .11 | .15 | .30 | .10 | .72 | .28 | .05 | .21 | (.84) | | | | |
| 14 | Work–family balance self-efficacy (T2) | 3.92 | 1.02 | .07 | .08 | .01 | −.09 | .02 | .06 | .12 | .20 | .28 | .51 | −.24 | .24 | .47 | (.88) | | | |
| 15 | Anticipated work interference with family (T2) | 4.51 | .97 | .04 | .03 | .10 | .02 | −.11 | −.10 | −.10 | −.08 | −.09 | −.34 | .62 | −.21 | −.12 | −.32 | (.73) | | |
| 16 | Employment attitude (T3) | 4.36 | 1.48 | .01 | .01 | −.14 | .02 | −.01 | .10 | .20 | .15 | .39 | .25 | .07 | .69 | .31 | .25 | −.13 | (.92) | |
| 17 | In-role performance (T3) | 4.25 | .81 | .14 | .09 | .15 | .14 | .00 | −.01 | .13 | −.02 | .14 | −.02 | −.03 | .12 | .15 | .26 | −.23 | .19 | (.86) |

*Note*: Scores in parenthesis are Cronbach's alpha. Education (0 = high school, 1 = college, 2 = undergraduate, 3 = postgraduate). Number of maternity leaves: maternity leaves that had been taken, including the current one and ranging from 1 to 3. Number of children: ranges from 1 to 3. $n = 100$ (control and T1/T2 measures), 81 (T3 self-rated measures), 72 (T3 supervisor-rated measures) and 64 (correlation between T3 self- and supervisor-rated measures). Correlations above .20 ($n = 100$), .22 ($n = 81$), .24 ($n = 72$) and .25 ($n = 64$) are significant at $p < .05$, and those above .26 ($n = 100$), .29 ($n = 81$), .30 ($n = 72$) and .32 ($n = 64$) are significant at $p < .01$.

**Table 4.** Results of random-intercept models in predicting outcomes at T3

| Analysis for in-role performance | Model 1 | | | | Model 2 | | | | Model 3 | | | | Model 4 | | | |
|---|---|---|---|---|---|---|---|---|---|---|---|---|---|---|---|---|
| | B | SE | p | Sig. | B | SE | p | Sig. | B | SE | p | Sig. | B | SE | p | Sig. |
| Age (years) (T1) | .05 | .03 | .10 | + | .05 | .03 | .11 | | .04 | .03 | .18 | | | | | |
| Education (T1) | .22 | .16 | .16 | | .22 | .16 | .18 | | .20 | .15 | .18 | | | | | |
| Number of maternity leaves (T1) | −.04 | .16 | .81 | | −.04 | .16 | .79 | | −.09 | .16 | .56 | | | | | |
| Proactive personality (T1) | .07 | .13 | .61 | | .07 | .13 | .61 | | .16 | .13 | .20 | | | | | |
| Family support (T1) | −.05 | .09 | .56 | | −.05 | .09 | .62 | | −.05 | .09 | .55 | | | | | |
| Managerial self-efficacy (T1) | .06 | .13 | .63 | | .06 | .13 | .63 | | .14 | .13 | .28 | | | | | |
| Managerial self-efficacy (T2) | .04 | .14 | .78 | | .04 | .14 | .76 | | −.12 | .15 | .42 | | | | | |
| Employment attitude (T1) | .03 | .08 | .69 | | .03 | .08 | .69 | | −.02 | .08 | .81 | | | | | |
| Work–family balance self-efficacy (T1) | | | | | −.01 | .09 | .90 | | −.14 | .09 | .13 | | −.14 | .09 | .10 | |
| Work–family balance self-efficacy (T2) | | | | | | | | | .33 | .12 | .01 | ** | .30 | .10 | .00 | ** |
| Intercept | 1.56 | 1.37 | .26 | | 1.60 | 1.40 | .25 | | 1.41 | 1.33 | .29 | | 3.61 | .37 | .00 | |
| -2loglikelihood | 165.68 | | | | 165.66 | | | | 158.14 | | | | 167.74 | | | |
| df | 8 | | | | 9 | | | | 10 | | | | 2 | | | |
| Δ − 2 loglikelihood | | | | | .02 | | | | 7.53 | | | | | | | |
| Chi-squared test for Δ-2loglikelihood | | | | | .90 | n.s. | | | .01 | ** | | | | | | |

| Analysis for employment attitude | Model 5 | | | | Model 6 | | | | Model 7 | | | | Model 8 | | | |
|---|---|---|---|---|---|---|---|---|---|---|---|---|---|---|---|---|
| | B | SE | p | Sig. | B | SE | p | Sig. | B | SE | p | Sig. | B | SE | p | Sig. |
| Age (years) (T1) | .00 | .03 | .90 | | .00 | .03 | .98 | | .00 | .03 | .98 | | | | | |
| Education (T1) | −.11 | .18 | .56 | | −.08 | .19 | .69 | | −.08 | .19 | .68 | | | | | |
| Number of maternity leaves (T1) | −.30 | .22 | .16 | | −.30 | .22 | .17 | | −.30 | .22 | .17 | | | | | |
| Proactive personality (T1) | −.19 | .16 | .24 | | −.20 | .16 | .21 | | −.20 | .16 | .21 | | | | | |
| Family support (T1) | −.08 | .10 | .40 | | −.10 | .10 | .31 | | −.10 | .10 | .31 | | | | | |
| Managerial self-efficacy (T1) | .10 | .16 | .52 | | .10 | .16 | .54 | | .10 | .16 | .54 | | | | | |
| Managerial self-efficacy (T2) | .29 | .16 | .08 | + | .28 | .16 | .09 | + | .28 | .18 | .12 | | | | | |
| Employment attitude (T1) | .77 | .10 | .00 | ** | .76 | .10 | .00 | ** | .76 | .10 | .00 | ** | | | | |
| Work–family balance self-efficacy (T1) | | | | | .08 | .11 | .44 | | .08 | .12 | .47 | | .21 | .15 | .15 | |
| Work–family balance self-efficacy (T2) | | | | | | | | | .00 | .15 | .99 | | .17 | .19 | .37 | |
| Intercept | 1.45 | 1.52 | .34 | | 1.18 | 1.55 | .45 | | 1.18 | 1.56 | .45 | | 2.95 | .73 | .00 | |
| -2loglikelihood | 231.92 | | | | 231.33 | | | | 231.33 | | | | 283.91 | | | |
| df | 8 | | | | 9 | | | | 10 | | | | 2 | | | |
| Δ − 2 loglikelihood | | | | | .59 | | | | .00 | | | | | | | |
| Chi-squared test for Δ-2loglikelihood | | | | | .44 | n.s. | | | .99 | n.s. | | | | | | |

Note: $n = 72$ for analysis for in-role performance, $n = 81$ for analysis of employment attitude. Education (0 = high school, 1 = college, 2 = undergraduate, 3 = postgraduate). Number of maternity leaves: maternity leaves that had been taken, including the current one and ranging from 1 to 3.
*$p < .05$;
**$p < .01$;
+$p < .10$.

better after they return to work. Work–family balance self-efficacy, however, did not help promote employment attitude. The null association between work–family balance self-efficacy and employment attitude may relate to how we measure employment attitude. Our measured employment attitude captures working mothers' intentions to keep working regardless of the potential impacts of

their work on their families; factors such as their beliefs on gender roles, financial situations and the family context of domestic work arrangements might determine its levels more than their work–family balance self-efficacy does. Meanwhile, our training program helps increase working mothers' managerial self-efficacy, which may inspire working mothers to develop a long-term career goal

beyond their current job positions. This speculation should be further examined.

Broadly speaking, our findings indicate that boosted work–family balance self-efficacy can serve as psychological resource that helps working mothers cope with the challenges in the transition from maternity leave to work, at least for their in-role performance. Unlike conventional family-friendly policies, such as maternity leave, which aim to support working mothers' adaptation to childbirth and new challenges in childcare, our study suggests that organizations can use our training program to address concerns about handling work–family balance issues to readapt to work after maternity leave. We believe that our training program supplements existing family-friendly policies and encourage organizations to use our training program to help those on maternity leave better prepare for eventual readaptation to work. It could be argued that having training during maternity leave will take time from the leave and prevent work mothers from detaching from work. However, the cost of taking the training during maternity leave may be paid off because the training boosts working mothers' psychological resources to cope with potential work–family balance issues. Working mothers can proactively prepare their return to work to avoid being overwhelmed and stressed out when they return to work. In addition, organizations can use the training as an opportunity to strengthen a positive social exchange with working mothers because it could create a positive loop between organizations' care for working mothers and working mothers' continuous contributions to the organization.

Nevertheless, our work has several limitations. First, the study was conducted in Japan, a country with unique social conditions around working mothers, such as a traditional distinction of gender roles, rigid social norms that encourage individuals to follow social expectations and widespread long working hours (Yoshikawa et al., 2018). Although we designed the training program to boost self-efficacy based on social cognition theory (Bandura, 1971, 1977, 2001), which is applicable in other societal contexts, the generalizability of our findings needs to be verified in further studies.

Second, owing to the practical challenges in recruiting participants who are taking maternity leave and ethical concerns about allocating participants to a control condition without giving them the training to help them return to work, we did not apply a random experimental design. Although we acknowledge that our research design might involve bias in participant selection, we have used a statistical approach to confirm that the participants signing up for our training program did not differ in their demographic characteristics from those who did not. Nevertheless, we acknowledge that more studies are needed to provide more evidence to support the effectiveness of our training program.

Third, we primarily focused on working mothers' self-efficacy in predicting their employment attitude and in-role performance. Although we did control for organization-level differences by adopting a mixed-effects model, this does not fully account for potentially relevant contextual factors, such as organizational climate, job design and supervisors' leadership. We chose this approach because our purpose in this study was to examine the effect of our training program. We recommend that researchers incorporate both individual and contextual factors to better understand working mothers' readaptation to work after maternity leave.

Finally, we only used employment attitude and in-role performance to examine working mothers' readaptation to work. The two outcomes only capture working mothers' and their supervisors' perspectives, respectively. We did not assess outcomes reflecting working mothers' work–family balance, such as work–family conflict, after they returned to work. We have also not included family members' perspectives to assess working mothers' readaptation to work. Future studies are encouraged to extend our study by considering a wide range of readaptation outcomes and including family members to access working mothers' readaptation.

**Open peer review.** To view the open peer review materials for this article, please visit http://doi.org/10.1017/gmh.2023.6.

**Data availability statement.** The data are not publicly available due to the agreement with the participating companies and participants.

**Author contribution.** The three authors contributed equally to this manuscript. Conceptualization: A.K., K.Y. and C.-H.W.; Data curation: A.K.; Formal analysis: K.Y.; Methodology: K.Y. and C.-H.W.; Writing – original draft: A.K., K.Y. and C.-H.W.; Writing – review and editing: A.K., K.Y. and C.-H.W.

**Financial support.** This research was supported by funding from the Japan Society for the Promotion of Science (Grant Number JP19K01864).

**Competing interest.** We would like to declare that one of the coauthors has interests that may potentially affect our work. The first author of this article is a cofounder of a training company, and the company offers training programs using the cases that we developed and tested in this study. She has a minority share of the company and delivers various training programs for the company on a fee basis. The two other authors do not have any interests, including employment, consultancies, stock ownership, honoraria, paid expert testimony, patent applications/registrations and grants or other funding, that may influence our work.

**Ethics standard.** The study was conducted in accordance with the approved guidelines and regulations from the Ethics Committee of the University of Shizuoka, to which the first author affiliates. Informed consent was obtained from all participants involved in the study.

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
