## [Reviewer Report]

Dear Editor,

We would like to submit the manuscript entitled “Facilitating transition from maternity leave to work for working mothers: A self-efficacy intervention study,” for consideration for publication in the Cambridge Prisms: Global Mental Health. We are submitting this paper to the Cambridge Prisms: Global Mental Health, as we believe it offers a high quality, original contribution, extending conversations that have been taking place within the journal.

Female workers face unique challenges in continuing and progressing their career, and as a result, gender inequality remains a critical challenge in many countries. Prior studies report transition from maternity leave to work is a critical career-transition period, and failure in readaptation often leads them to quit professional career. We address the issue by developing and deliverying a theory-based training program, drawing on self-efficacy theory. Analysis of a longitudinal dataset that we collected from the participants and their supervisors suggest that the participants increased work–life balance self-efficacy and managerial self-efficacy after the training and that the two types of self-efficacy predicted their career attitudes and work behaivors 6 months after the participants returned to work. These results suggest the newly developed training can positively influence the psychological functioning of working mothers and thus, supplements existing family-responsive practices in facilitating female workers' career progress.

We would like to refrain from making the dataset available because the data is collected upon a condition that we do not disclose information to third parties. We confirm that our research follows ethical guidelines of institutions that authors affiliate to, and this manuscript represents results of original work that has not been published nor is currently being considered for publication elsewhere.

Yours sincerely, 

Author team

Akiko Kokubo

University of Shizuoka 

akokubo@u-shizuoka-ken.ac.jp

Katsuhiko Yoshikawa

Shizenkan University

Email: katsuhiko78@gmail.com

Chia-Huei Wu

University of Leeds 

China Medical University, Taichung, Taiwan

Email: chiahuei.wu@gmail.com

---

## [Reviewer Report]

*Comments to Author*: This manuscript described a theory-based intervention with a Japanese sample of female maternity leave employees and investigated the intervention’s effect on working-family balance self-efficacy, managerial self-efficacy, and other job-related outcomes. Overall, I found this to be a clear and well-written manuscript with rigorous methods for three waves data collection (about one year period) and describing an important finding in this research area. I also want to note the non-WEIRD sample, especially in the Japanese context, as another strength of this manuscript (however, in the current manuscript, the authors have illustrated this as a limitation). I have only a few suggestions/questions :

1. Although the results (see Table 4) demonstrated that working-family balance self-efficacy did not predict career commitment. However, the zero correlations (see Table 2) showed that both Time 1 and Time 2 working-family balance self-efficacy correlated with career commitment (r = .25). Therefore, there is still a direct effect of working-family balance self-efficacy but decreased as other controlled variables included in the model simultaneously. So, it will be better to demonstrate step by step for these models (with and without covariates), and the wording in the discussion section could be relative positively if the direct effect remained under some conditions. In addition, the authors could also treat different controlled variables more specifically because there might not need to include all the covariates in the final model. (Wysocki, Lawson, & Rhemtulla, 2022).

Wysocki, A. C., Lawson, K. M., & Rhemtulla, M. (2022). Statistical control requires causal justification. Advances in Methods and Practices in Psychological Science, 5(2), 1-19. https://doi.org/10.1177/25152459221095823

2. Corresponding with the previous suggestion, the outcome variables in the current study were correlated with moderate to high effect sizes (see Table 2). It could be integrated with their similar features or their theoretical background. However, it is still okay if the authors treated these outcomes independently.

Minor issues:

3. Please add a footnote in Table 3 about the significant level at p < .05 (maybe r = .20) of the sample size at Time 1, 2 and 3.

Overall, this was an excellent paper, and I applaud the authors’ attempts to investigate the intervention study on maternity leave for female employees in East Asian society. I hope my comments will facilitate the future progress of this research.

---

## [Reviewer Report]

*Comments to Author*: This interesting examined the impact of a training intervention on working mothers returning from materiality leave. I appreciated the longitudinal design of the study. However, some issues concerning hypothesis development require further elaboration. I provide the following comments.

1. The theoretical foundation of the current study requires more elaboration. Although self-efficacies are the focus of the intervention program, why these self-efficacies are used to predict these outcome variables remain unclear. For example, based on the purpose of the study, I expect work-family conflict to be a meaningful outcome variable for work-family balance self-efficacy. However, this construct was not examined (although the anticipated work-family conflict was collected). It would be necessary to offer a clear theoretical justification for why these outcomes are chosen. Finally, the reason why the work-family conflict was not examined should be addressed as a potential limitation of the study.

2. Based on the current research design, the author should reframe the theoretical framework of the JD-R model and treat self-efficacies as important and trainable personal resources. By doing so, you can argue self-efficacies as predictors of work behaviors (both in-role and extra-role) and job crafting-related behaviors concerning job demands.

3. I have concerns about the measure used for career commitment. These items reflect more of work-family conflict rather than career commitment (Blau, 1985; e.g, I would take any job paying the same). I suggest you use another name rather than career commitment, as the construct reflects turnover intention due to work-family conflict rather than career commitment.

4. The reason why examining two types of self-efficacies at the same is a theoretical contribution needs to be specified. In the current manuscript, no relevant hypotheses (e.g., the relative importance of the two self-efficacies) were proposed. Because the two types of self-efficacies are merely control variables for each other, the examination does not constitute a theoretical contribution.

5. Following point 4, I have concerns for Hypotheses 2 and 3. Since both self-efficacies are related to work context and no clear justification is provided, it is unreasonable to propose a particular type of self-efficacy to be only related to specific outcomes. Furthermore, available meta-analyses (e,.g., Liao et al., 2019) also suggest that work-family constructs are related to work and career attitudes and behaviors. Consider revising your hypothesis concerning the relative impacts of the two self-efficacies. For example, the relationship between post-training work-family balance self-efficacy in the following outcomes: (a) career commitment, (b) decreasing hindering job demand, and (c) in-role performance would be stronger than the relationship between post-training managerial self-efficacy and the above outcome variables.

6. One potential threat to the internal validity of the current study is the social support gained from participating in the intervention session. In the present research context (Japan), workplace social support is a valuable but scarce social resource for employees. Participation in this training program helps working mothers build social resources, providing them with instrumental and emotional support. Furthermore, social resources may also help them generate personal resources such as self-efficiencies in the current study context (Hobfoll et al., 1990). Please discuss this as both a potential limitation and a future research direction in the discussion.

7. Table 4 requires some clarification. I suggest you refer to the APA format. (1) For example, when control variables are collected should also be specified. (2) Including a significance column under each outcome variable or the p-values at the bottom of the table is unnecessary. (3) “Change by the inclusion of T2 variables” needs to be clarified. Please report the R-sq and R-sq change for the following three hierarchical regression blocks (control only, control + T1; Control +T1+T2). (4) Please rearrange the order of columns such related constructs are listed next to each other: career-related attitudes (M1 and M4), work performance (Model 2 and Model 5 in-role, extra-role), and job crafting-related variables (Model 3 and Model 6).

8. Please move the Development of the Training Program to the method session. Please provide more information about the procedure. For example, did all participants start the first session (T1) before returning to the workplace? What are the time lags between T1 and T2? Did all participants return to their workplace at T2?

9. For Table 1, please specify the anchor of tenure, education, and the number of the material leave in the note. Please also determine the time control variables are measured.

10. Please report the t-test results(means, SD for each group) on page 14.

Reference

Hobfoll, S. E., Freedy, J., Lane, C., & Geller, P. (1990). Conservation of social resources: Social support resource theory. Journal of Social and Personal Relationships, 7(4), 465-478.

Liao, E. Y., Lau, V. P., Hui, R. T.-y., & Kong, K. H. (2019). A resource-based perspective on work-family conflict: Meta-analytical findings. The Career Development International, 24(1), 37–73. https://doi.org/10.1108/CDI-12-2017-0236

---

## [Reviewer Report]

*Comments to Author*: This is an interesting article, describing a relevant topic. It describes a preliminary pilot investigation of a self-efficacy intervention to help new mother in their return to work in Japan. 

However, the methodology has important limitations. In general, the efficacy or effectiveness of an intervention can be examined using a randomized controlled trials (RCTs). Since this study did not include a control arm, any conclusions regarding the effects are preliminary. This should be clearly stressed, in the abstract, and in the discussion (where it is only clearly touched upon).

Further, it is important that the authors include a flow chart (see: https://www.consort-statement.org/consort-statement/flow-diagram).

A large number of outcomes was included, but what was considered to be the main outcome measure?

And were any checks performed on how the programme was delivered?

The authors should address these concerns in a major revision.

---

## [Reviewer Report]

We have uploaded our response letter as a supplementary material. Please see it for our responses.